

# Assessment of Snow, Sea Ice, and Related Climate Processes in Canada's Earth-System Model and Climate Prediction System

Paul J. Kushner[1], Lawrence R. Mudryk[2], William Merryfield[2], Jaison T. Ambadan[3], Aaron Berg[3], Adéline Bichet[4], Ross Brown[2], Christopher P. Derksen[2], Stephen J. Déry[5], Arlan Dirkson[6], Greg Flato[2], Christopher G. Fletcher[7], John C. Fyfe[2], Nathan Gillett[2], Christian Haas[8,9], Stephen Howell[2], Frédéric Laliberté[2], Kelly McCusker[10], Michael Sigmond[2], Reinel Sospedra-Alfonso[2], Neil F. Tandon[2], Chad Thackeray[7], Bruno Tremblay[11], Francis W. Zwiers[12]

[1]Department of Physics, University of Toronto, M5S 1A7, Canada
[2]Climate Research Division, Environment and Climate Change Canada, M3H 5T4, Canada
[3]Department of Geography, University of Guelph, N1G 2W1, Canada
[4]CNRS-LGGE/MEOM, 38041 Grenoble, France
[5]Department of Environmental Science, University of Northern British Columbia, V2N 4Z9, Canada
[6]School of Earth and Ocean Sciences, University of Victoria, V8W 2Y2, Canada
[7]Department of Geography and Environmental Management, University of Waterloo, N2L 3G1, Canada
[8]YDepartment of Earth and Space Science and Engineering, York University, M3J 1P3, Canada
[9]Climate Sciences Division, Alfred Wegener Institute, 27570, Germany
[10]Department of Atmospheric Sciences, University of Washington, 98195-1640, United States
[11]Department of Atmospheric and Oceanic Sciences, McGill University, H3A 0B9, Canada
[12]Pacific Climate Impacts Consortium, University of Victoria, V8P 5C2, Canada

*Correspondence to*: Paul J. Kushner (paul.kushner@utoronto.ca)

**Abstract.** This study assesses the ability of the Canadian Seasonal to Interannual Prediction System (CanSIPS) and the Canadian Earth-system Model 2 (CanESM2) to predict and simulate snow and sea ice from seasonal to multi-decadal timescales, with a focus on the Canadian sector. To account for observational uncertainty, model structural uncertainty, and internal climate variability, the analysis uses multi-source observations, multiple Earth-System Models (ESMs) in Phase 5 of the Coupled Model Intercomparison Project (CMIP5) archive, and initial condition ensembles of CanESM2 and other models. It is found that the ability of the CanESM2 simulation to capture snow-related climate parameters, such as cold-region temperature and precipitation, lies within the range of currently available international models. Accounting for the considerable disagreement among satellite-era observational datasets on the distribution of snow water equivalent, CanESM2 has too much springtime snow cover over the Canadian land mass, reflecting a broader Northern Hemisphere positive bias. It also exhibits retreat of springtime snow generally greater than observational estimates, after accounting for observational uncertainty and internal variability. Sea ice is biased low in the Canadian Arctic, which makes it difficult to assess the realism of long-term sea-ice trends there. The strengths and weaknesses of the modeling system need to be understood as a practical tradeoff: the Canadian models are relatively inexpensive computationally because of their moderate resolution, thus enabling their use in operational seasonal prediction and for generating large ensembles of multidecadal simulations. Improvements in climate prediction systems like CanSIPS rely not just on simulation quality but also on using novel





observational constraints and the ready transfer of research to an operational setting. Improvements in seasonal forecasting practice arising from recent research include accurate initialization of snow and frozen soil, accounting for observational uncertainty in forecast verification, and sea-ice thickness initialization using statistical predictors available in real time.

## 1 Introduction

Seasonal snow cover and sea ice are integral to the cultural identity, history, and economy of northern nations like Canada. They also exert an enormous physical influence on the earth system, ranging from local interactions with winds and temperatures in the Arctic and snow-covered regions, to larger-scale interactions with weather systems and ocean circulation, to global-scale influences on the Earth's energy balance. In recent decades, dramatic changes in Canada's snow cover and sea ice have been witnessed and documented (Derksen et al., 2012; Najafi et al., 2015). This has driven the need to

better understand and predict these fields for the coming seasons, years and decades. To address this need, Canada has helped lead the global effort to better observe and model snow, sea ice, and related climate parameters. This effort includes Canadian contributions to the International Polar Year (e.g. Kulkarni et al., 2012), to the development of leading earth system model and climate prediction systems (Merryfield et al., 2013a; Sigmond et al., 2013; van den Hurk et al., 2016), and to leadership of ongoing field and remote sensing efforts (King et al., 2015).

As part of Canada's larger effort in snow and sea-ice research, this study is focused on seasonal and longer timescale prediction of terrestrial snow, sea-ice cover, and related climate variability. The purpose of the study is to evaluate the ability of Canada's current earth system model (ESM) and climate prediction system to carry out this kind of prediction in the context of the development of new observational products. This work was undertaken by the Canadian Sea Ice and Snow

Evolution Network (CanSISE), a core project of the Climate Change and Atmospheric Research Initiative of the Natural Sciences and Engineering Research Council of Canada (CCAR/NSERC)[1]. Model evaluation, which typically compares a model to observations, needs to account for several sources of uncertainty, including impacts of spatial and temporal sampling in the presence of internal climate variability and observational uncertainty (whether instrumental error or errors related to data processing and retrieval systems). Our evaluation of Canadian models is helped by ready access and

comparison with output from internationally available models, to provide a suitable scientific context.

---

[1] The CanSISE Network was funded for five years starting in 2013. It is a partnership between several Canadian Universities (Toronto, British Columbia, Guelph, McGill, Northern British Columbia, Victoria, Waterloo, and York); Environment and Climate Change Canada [ECCC; research groups include the Canadian Centre for Climate Modelling and Analysis (CCCma) and the Climate Processes Section, both in the Climate Research Division; and the Canadian Ice Service (CIS)], and the Pacific Climate Impacts Consortium (PCIC).





This study focuses on snow, sea ice and related climate parameters relevant to the Canadian land mass and the pan-Arctic region. The Canadian ESM and climate prediction system has been studied in a variety of related settings (e.g. Arora et al. 2011; Merryfield et al. 2013a, b; Gillet et al. 2012; Sigmond et al. 2013, Kirtman et al. 2013; Flato et al. 2013). We here seek to more fully assess simulation and prediction of seasonal snow cover and regional sea-ice variability accompanied by a

more complete a characterization of observational uncertain, model structural uncertainty, and internal climate variability. After reviewing the current generation Canadian Seasonal to Interannual Prediction System and Canadian Earth-System Model 2 (CanSIPS and CanESM2; section 2), we characterize climatological behavior and trends for snow and sea ice in these systems (section 3), provide an overview of recent developments in seasonal snow and sea-ice prediction (section 4), and conclude (section 5) with a summary and discussion of new directions for prediction system development.

**2 Models and data used**

CanSIPS and its component models provide Environment and Climate Change Canada's (ECCC) current operational forecasts for climate variability on seasonal to interannual (several-month to multiple-year) timescales. CanSIPS (Merryfield et al. 2013a) combines a dynamical ocean model (that simulates three-dimensional ocean circulation and heat and salinity transport) with atmosphere, land surface (including snow) and sea-ice component models in a coupled framework in which

all model components interact. The system's surface energy and water budgets are in sufficient balance to avoid climate state drift over the course of longer simulations. It includes 1) a data assimilation system that estimates realistic initial states of the atmosphere, ocean, land, and sea ice to start the forecasts; 2) two separate coupled climate models (the earlier generation Canadian Coupled Model 3 [CanCM3] and the later generation Canadian Coupled Model [CanCM4]) that advance the simulated system from this initial condition; and 3) diagnostic systems to analyze the output and generate useful forecasts for

operational use within ECCC's Meteorological Service of Canada. Evaluations of CanSIPS need to consider all three parts of the seasonal prediction system.

The coupled climate models and ESMs can also be run independently of the data assimilation system. Run in this way, the same modelling system can be used to project long-term climate behavior under the influence of greenhouse gas emissions

and other anthropogenic and natural forcings, but independently of particular initial conditions in the atmosphere and ocean. ECCC's Canadian Centre for Climate Modeling and Analysis (CCCma) uses an extension of CanCM4 called the Canadian Earth-System Model 2 (CanESM2; Arora et al., 2011; Scinocca et al., 2016), which includes interactive land and ocean carbon cycle components to project the future state of global temperature, circulation, carbon dioxide concentrations, etc. under the influence of external forcing.

As with most other models participating in CMIP5, CanESM2 does not use flux adjustments that artificially constrain the climate system to be in a state of energy and water balance; CanCM4 and CanESM2 use time varying volcanic forcing, and





include a prognostic (interactive) carbon cycle that uses biological models to simulate carbon cycling in the coupled atmosphere/land/ocean/biosphere system. These systems compare well to other earth system models and climate prediction systems (Merryfield et al., 2013a and other citations in the introduction). CanCM3 has a very small annual mean sea surface temperature bias, and CanCM4 reduces the global mean absolute error of ocean surface temperatures compared to CanCM3,

indicating an overall improvement in the coupled ocean atmosphere state captured in the latest generation model (Merryfield et al., 2013a). Relative to CanCM3 and observations, CanCM4 tends to warm more rapidly under the effects of anthropogenic climate change over the 1970-2009 period. This characteristic is relevant to snow and sea-ice variability and trends and variability in CanESM2 (section 3). In CanCM3, the simulation is characterized by excessive pan-Arctic sea-ice cover in summer and winter and a small rate of sea-ice loss compared to observations. In CanCM4, while there is still

excessive sea-ice cover in winter, there is too little sea ice in summer (section 3). The rate of sea-ice loss in CanCM4 is more in line with recent observations than that in CanCM3 (Stroeve et al., 2012); however, caution is required to interpret recent sea-ice loss rates in light of the large amount of multidecadal variability expected in these trends (e.g. Notz, 2012; Swart et al., 2015). CanSIPS, combining CanCM3 and CanCM4, is able to show multi-month skill in seasonal forecasts of detrended sea-ice area anomalies, comparable to that obtained in other modelling systems (Merryfield et al., 2013b), and generally

enhanced skill relative to a statistical persistence forecast (Sigmond et al., 2013). The assessed skill depends on the verification dataset (Sigmond et al. 2013), especially for total (non-detrended) anomalies. Such issues will be revisited in this study.

The CanSIPS and CanESM2 systems have moderate spatial resolution compared to many other CMIP5 models

(approximately 2.8° horizontal grid spacing and up to 35 vertical levels in the atmosphere; approximately 100 km horizontal grid spacing and up to 40 levels in the ocean). This resolution accounts for constraints on advanced computing resources, sufficiently resolving salient features of the global atmosphere-ocean circulation, while still permitting the execution of large ensembles of model simulations to adequately sample internal variability under different external forcings. ECCC has also made a complementary multi-year investment in regional climate modelling (Scinocca et al., 2016) to provide much higher

resolution over Canada to address the shortcomings of coarse resolution.

Our assessment of CanSIPS/CanESM is enhanced by two recent research products arising from CanSISE: the Blended-5 snow water equivalent (SWE) dataset of Mudryk et al. (2015) and the CanESM2 Large Ensemble of simulations from CanESM2. The Blended-5 dataset addresses the need for a SWE verification dataset, and, potentially, for initialization of

snow-related parameters in CanSIPS and other prediction systems. Blended-5 builds on long-term work of ECCC (e.g. Brown et al., 2010; Derksen and Brown, 2012; Brown and Derksen, 2013) and consists of an ensemble of gridded SWE datasets over 1981-2010 from a variety of sources including remote sensing, land surface assimilation systems, and reanalysis driven snow models.





The CanESM2 Large Ensemble (e.g. Sigmond and Fyfe, 2016) consists of four sets of 50 simulations each of CanESM2 that examine the impact of natural and anthropogenic forcings over the period 1951-2100 in the presence of internal climate variability. During the period 1951-2005, the large ensemble is run using CMIP5 historical forcings (Taylor et al., 2012); from 2006-2100, the RCP8.5 CMIP5 scenario is used. The first ensemble set, which applies all available external forcings,

will be the one used here. Additional sets of attribution integrations include just historic natural external forcings (solar and volcanic), just historic anthropogenic aerosol forcings, and just stratospheric ozone forcing. Each realization in each set is identical apart from its initial conditions. The CanESM2 Large Ensemble affords a more thorough assessment of the uncertainty connected to internal climate variability than is possible in past analysis efforts and has been used in several current and ongoing studies (Sigmond and Fyfe, 2016; McKusker et al., 2016; Gagné et al., 2016; Fyfe et al., 2016; Mudryk

et al., 2017; Kirchmeier-Young et al., 2016). We also use similar initial condition ensembles of the National Center for Atmospheric Research Community Earth-System Model 1 (NCAR CESM1; Kay et al., 2015) and the NCAR Community Climate System Model 4 (CCSM4; Mudryk et al., 2013). Other observational sources and modelling results used in this study will be described in the text and figure captions.

**3 CanESM2 climatology and trends**

We first evaluate the climatological characteristics of CanESM2 temperature, precipitation, and snow water equivalent (SWE), taking CanESM2 to be representative of the physical coupled model CanCM4, which is one of the component models of CanSIPS. In winter and spring, the distribution of surface temperature over Canada is well reproduced in CanESM2, although a warm bias is evident in both seasons (left and central panels of the top rows of Figs. 1-2). The Taylor (2001) diagram for temperature (top row, right panels of Figs. 1-2) shows that CanESM2 compares well to other CMIP5

models (the same models and realizations are used in Mudryk et al., 2017) in capturing the spatial pattern and correlation with observations, although the spatial gradients are somewhat stronger than observed for winter (as shown by the distance of the CanESM2 point from the origin in the Taylor diagram). In precipitation (central row of Figs. 1-2), the general pattern and spatial gradient strength are captured in the model, but there is excessive precipitation over the Western Cordillera, sub-Arctic and Arctic in both seasons. This excessive precipitation contributes towards a bias of excessive SWE over much of

Western Canada and the Canadian Sub-Arctic that is particularly pronounced in spring (lower row of Figs. 1-2). CanESM2 SWE has greater spatial variance than the Blended-5 SWE ensemble mean and most of the individual component datasets of the Blended 5 (lower right panel of Figs. 1-2). Generally speaking, the Taylor diagrams in Figs. 1-2 suggest that CanESM2 is well within the state of the art of current models for the climate parameters related to seasonal snow cover.

Observed SWE climatology, variability and trends are relatively non-robust compared to variables such as temperature (Mudryk et al., 2015, 2017) and for this reason we assess some aspects of the spread across the Blended-5 SWE datasets. Individual observational datasets contributing to Blended-5 also show stronger spatial gradients than the Blended-5 mean





(circles filled with light brown in the Taylor diagram in the lower right panels of Figs. 1-2). This is in part expected because the observational mean will cancel random errors. However, this also suggests that there is considerable uncertainty in the spatial variance, and so it is difficult to assess how realistically spatial variance is captured in CanESM2 and the other CMIP5 models. This observational uncertainty is also strongly evident in the seasonal cycle of total snow mass aggregated

for Canada and the Northern Hemisphere, as well as geographic subregions (Fig. 3). There is a large spread in the snow products for total snow mass for most regions (gray shading), with the NASA Global Land Data Assimilation System (GLDAS) providing an estimate well below the multi-dataset mean, the MERRA reanalysis dataset typically providing a central estimate and the maximum estimate varying with region among the remaining three datasets (individual datasets not shown here but are discussed in Mudryk et al. 2015).

While there is considerable observational uncertainty, it is nevertheless possible to assess the realism of CanESM2's SWE simulation. The positive bias of CanESM2 relative to the observational mean (Figs. 1-2) is evident in the seasonal cycle in snow mass over Canada (bottom row of Fig. 3), especially in spring, and reflects a broader Northern Hemisphere positive bias (top row of Fig. 3). The CanESM2 snow climatology is plotted as originally available on the model's land grid (dark

teal points in Fig. 3) and as adjusted to reflect the observational mask which is on a finer scale (light teal points). The adjustment is downward because some of the model's snow mass is located in grid cells that, in reality, are only partially covered by land. CanESM2's SWE is generally within the observational range but is above that range in springtime. This springtime positive bias is exacerbated over Canada in particular (bottom row), especially outside the Arctic. For comparison, the CMIP5 multi-model mean (shown, for Canada only, in the bottom row) does not feature such a bias. The

CMIP5 model range (not shown) spans from the lowest observational estimate to above the CanESM2, but CanESM2 is on the high end, especially during spring in the midlatitudes. Our assessment is that, especially in midlatitudes, CanESM2 simulates excessive springtime snow associated with excessive wintertime precipitation building up throughout winter and into spring (middle rows of Figs. 1-2).

A standard target for snow process analysis in climate models is snow-cover extent trends, which are strongly temperature controlled (e.g. Brutel-Vuilmet et al., 2013; Mudryk et al., 2017). Assessing the ability of models to capture these trends needs to account for natural variability, forced variability, observational uncertainty, and intermodel differences. We show in Fig. 4 the trends in snow cover extent (SCE) derived from the Blended-5 dataset (by converting SWE to SCE using a threshold of 4 mm; this threshold was tested in Mudryk et al., 2017) for the Northern Hemisphere in January-March and

April-June. In both seasons, there is a spread of observed seasonal snow cover reduction estimates from 0.0 to -0.5 million $km^2$ per decade in winter (based on a simple interquartile range for this small number of observational datasets), and from -0.2 to -0.6 million $km^2$ in spring. The red horizontal line in the box plot represents the median over the Blended-5 datasets. The range of trends from the CanESM2 Large Ensemble, NCAR CESM1, and NCAR CCSM4 suggests that internal variability alone provides an uncertainty range of about 0.5 million $km^2$ per decade. Assuming internal variability is well





represented in the models, this is the limit of precision we can expect in assessing recent trends. CanESM2 consistently produces greater snow loss than NCAR CCSM4 and CESM1, especially in AMJ. We conclude that all the models displayed fall within wintertime snow retreat estimates, that NCAR CCSM4 and CESM1 overlap with estimates of observed snow retreat in spring, but that CanESM2 exhibits more spring snow retreat than our best estimate of the observations. This

excessive snow retreat is associated in part with excessive global warming in the model mentioned in section 2 (Mudryk et al., 2017).

The anomalous wintertime 1981-2005 temperature trend pattern (i.e. the trend with the spatial mean removed, upper left panel of Fig. 5) represents a predominantly positive meridional gradient of temperature change from southern to northern

Canada, reflecting hemispheric wintertime polar amplification of warming. The same field in individual realizations of CESM1 and CanESM2 has a spatial correlation in the range of -0.6 to +0.6 (Taylor diagram in lower left panel of Fig. 5), suggesting that these patterns are affected by significant internal variability (Deser et al., 2012). There are more realizations with positive than negative spatial correlation in winter temperature trend patterns, which is consistent with the anticipated effect of anthropogenic forcing. Wintertime temperature trends systematically show greater spatial variance than the

estimated warming pattern from the single observational temperature dataset employed here. This could be related to stronger (more negative) meridional gradients in temperature and its trends in the models compared to the observational temperature dataset. Spring time temperature trend patterns (top left panel of Fig. 6) feature anomalously negative changes in the Canadian prairie regions and positive changes around the coastal regions. It is harder to find realizations in the springtime season that correspond to the observed pattern (left Taylor diagram in bottom row of Fig. 6), and spatial variance

of the temperature trends appear to be biased high as in winter. For precipitation, there is little evidence of consistent pattern matching between the observations and individual realizations, for winter or spring (middle column, top and bottom panels of Figs. 5-6). For SWE spatial patterns (right column, top and bottom panels of Figs. 5-6), the structural details of the trend maps are also not readily found in the models compared to the mean of the Blended-5 SWE. Compared to the ensemble mean of the Blended-5 observations, the simulations show greater spatial variance in SWE trends, but this is partially owing

to smoothing of spatial structure in observational errors, as is shown by the scatter of SWE trends by individual contributors to the Blended-5 dataset (light brown circles in Taylor diagrams).

It is possible to find individual realizations that, by one measure, either match fairly well or fairly poorly with observed trends (second and third rows of Figs. 5-6). The best-match realization (second row of Figs. 5-6) exhibits tradeoffs across

fields, for example in the ability to represent the structured pattern of springtime precipitation change (r=0.38 for middle panel of second row of Fig. 6) versus wintertime temperature change (r=0.01 for left panel of second row of Fig. 5). The worst-match case (third row of Figs. 5-6) exhibits a similar range of correlations, on the negative side, and generally looks quite different from the best-match case. This preliminary analysis of intraensemble variability suggests limits on how much regional-scale information about changes for snow cover and related climate variables can be extracted from ESMs. This



reflects the basic point that caution is needed in judging a model on its ability to reproduce spatial patterns of trends in SWE and related climate parameters, even on these multidecadal timescales.

The spatial pattern of CanESM2 temperature and precipitation trends is generally representative of that found in individual
realizations of the CMIP5 datasets, in the sense that the individual realizations of CanESM2 and other CMIP5 models have positive pattern correlations with the CMIP5 multi-model mean (Taylor diagrams not shown). Consistently, the CMIP5 multimodel mean of the temperature and precipitation trends are generally similar to the CanESM2 ensemble mean (winter example shown in the top two rows of Fig. 7). However, for SWE, we find CanESM2's pattern is typically opposite that of individual realizations from other models in CMIP5 (not shown) as is also evident in the ensemble mean (bottom row of Fig.
7). In particular, CanESM2 shows a strong positive trend in the Western Cordillera and a weaker positive trend in Southern Ontario and eastern Canada in both winter (Fig. 7) and spring (not shown), whereas a reduction of SWE is found in these regions and seasons in CMIP5.

Turning to sea ice, we recall that summertime sea-ice area or extent is biased low in CanESM2 (Stroeve et al., 2012;
Merryfield et al. 2013a; Laliberté et al. 2016), which is borne out in the Canadian Arctic sector (top two panels of Fig. 8), where CanESM2 has less than half of the observed sea-ice coverage in the Beaufort Sea-Arctic Ocean sector. Further limiting the utility of regional sea-ice analysis with this model is the moderate spatial resolution of the model and its associated land-sea distribution, particularly in the Canadian Arctic Archipelago (central panel of Fig. 8). The summertime sea-ice extent is among the lowest of all CMIP5 models in the Canadian Arctic as a whole. In Canadian Arctic regions,
summertime sea-ice extent is biased low in the Beaufort Sea and is practically zero in Hudson Bay and Baffin Bay (Fig. 9, left column). This bias contributes to the outcome that the sea ice reaches nominally ice-free summertime conditions at times comparable to present-day in CanESM2. The bias is evident throughout the seasonal cycle in most regions (Fig. 9, right column), with the exception of Baffin Bay, although not as extreme relative to other models in other seasons as it is in summer. In this respect, the quality of simulation in CanESM2 is not as good as that of other ESMs such as NCAR CESM1
(lower panel of Fig. 8), which provide a better baseline for regional sea-ice studies both in terms climatology and land-sea distribution. Process investigations of sea ice by CanSISE include a focus on the relationship between sea-ice drift and Arctic winds, since realistic sea-ice dynamics are crucial for accurate representation of sea ice (Notz, 2012). International Arctic Buoy (IABP) Programme measurements (Tschudi et al., 2016) show that sea-ice drift speed peaks in September, when sea ice is thinnest (lower panel of Fig. 10), and not at the time of peak wind speed in December. However, in
CanESM2, the peak sea-ice drift speed occurs in November and is more in phase with the seasonal cycle of near-surface wind speed. Other models in the CMIP5 archive that have more modern sea-ice components are able to reproduce more closely the observed seasonal cycle of sea-ice drift speed (Tandon et. al., submitted manuscript). These results provide strong motivation to transition to a modeling system with improved sea-ice and related processes in the Arctic.





### 4 Snow and sea-ice related forecast performance and development of CanSIPS

Operational seasonal forecasts based on coupled global ocean-atmosphere models have been produced for about two decades internationally (Stockdale et al., 1998) and in Canada (by CanSIPS) since 2011. Over this period the main emphasis has been on predicting seasonal meteorological variables describing near-surface temperature, atmospheric circulation and
precipitation, as well as sea-surface temperatures which are a major driver of seasonal climate variations. Potential has also existed for such systems to usefully predict additional variables, including snow and sea ice, particularly as the sophistication of the models and the methods used to initialize them have increased. With respect to the cryosphere, however, such capabilities have received little attention until recently.

Research carried out under CanSISE examined the ability of CanSIPS both to realistically initialize SWE and to predict future SWE variations (Sospedra-Alfonso et al., 2016a, b, c). This was the first study of snow in an operational seasonal forecast system. Regarding seasonal prediction of snow by CanSIPS, anomaly correlation skill for wintertime SWE is high at short lead times, and remains statistically significant (greater than 0.3) at lead times of at least 6 months for certain regions (Fig. 11), which suggests potential for practical utilization of such forecasts. Two primary sources of potential predictability
(PP, defined as the ratio of 'signal' variance describing interannual variability of ensemble means to total variance consisting of the sum of 'signal' and 'noise' components) and skill in CanSIPS forecasts of SWE have been identified (Sospedra-Alfonso et al., 2016b, c). The first, which is most important at short lead times, is the demonstrated ability of CanSIPS to provide realistic initial values for SWE (Sospedra-Alfonso et al., 2016a), combined with the natural tendency for SWE anomalies to persist throughout the snow season in regions where winter temperatures remain below freezing, so that the
snowpack accumulates until the onset of spring melt. The second main source of PP and skill, which becomes increasingly prevalent at longer lead times, is the ability of CanSIPS to predict future climatic conditions such as temperature and precipitation anomalies which influence snow accumulation and melt. A large part of this type of predictability and skill arises from ENSO, which strongly influences winter climate in North America and is skillfully forecast by CanSIPS up to a year in advance.

The value of skillful seasonal forecasting of snow in turn depends on process representation at the land-surface. For example, Ambadan et al. (2015) have investigated the impact of initialization of SWE, soil liquid water, and soil frozen water on potential predictability of springtime surface air temperature in the CanSIPS system (Fig. 12). Realistic initialization of these variables enhances potential predictability by as much as 30% in terms of variance explained within the
potential predictability framework. This shows that it is important to regard snow initialization in the broader setting of land surface initialization, and that there is evidence for quantitative improvement in regional predictability as more observational information on the state of the land surface is brought into the prediction system. Current operational practice in CanSIPS uses observed atmospheric forcing to bring the land surface (including soil moisture and snow cover) into a realistic state.





Although this procedure performs reasonably well for snow (within observational uncertainty), potential remains for improving the initialization and forecasting of snow and other land variables by assimilating observation-based land data directly in real time.

Blending different sources of data from highly uncertain observations has led to improved characterization of the forecast skill of the CanSIPS system. Fig. 13 shows the degree of agreement between SWE forecasts from CanSIPS and several SWE products over Canada (similar results are found for other regions). The degree of agreement is measured as the anomaly correlation coefficient for a one-month forecast (with lead 0 from initialization). The five datasets are the Blended-5 dataset (blue) and four individual datasets including two components of the Blended-5 dataset. Even though all observational

datasets are being compared to the same forecast, it is the Blended-5 dataset, capturing the mean of several observational datasets, that agrees best with the forecast. It is clear that improving verification datasets through blending, which can be reasonably expected to lead to cancellation of independent errors in observational estimates, impacts assessed agreement with the forecast. To reiterate, in this case, improved calculated skill is derived from an apparent improvement in the quality of the verification data and not an improvement of the forecast (Sospedra-Alfonso et al., 2016b). Whether or not such

improved consistency might be found for the prediction of other quantities, the broader point is that there is a need to ensure that verification data is continually updated in order to fairly compare predictions to the best available data (Massonet et al., 2016).

Recent research in snow analysis and observational datasets is expected to support operational improvements in CanSIPS

and hence in ECCC's operational prediction capacity. For example, CanSISE work has led to new efforts to develop an operational real-time snow amount forecast for the coming months, which could be used in several impacted sectors such as outdoor recreation, water resource planning, and agriculture (Fig. 14, snow amount forecast shown as above and below normal SWE amounts). In this successful proof of concept, we note satisfactory general agreement with the MERRA analysis, which is independent of CanSIPS and is itself subject to some uncertainty. This indicates promise for this new

forecast product, while highlighting issues of observational uncertainty addressed in part by our recent research.
Much as for snow, the ability of global climate model-based seasonal forecasting systems to predict sea ice has also received little attention until recently, although such assessments have now been carried out for several systems. In the area of sea-ice prediction, CanSIPS hindcasts, despite some of the simulation deficiencies described above, have demonstrated skill in seasonal predictions of sea ice (e.g. Sigmond et al., 2013; Merryfield et al., 2013b). While these prior studies have focused

on forecast skill of area-integrated quantities such as sea-ice area, recent work (Sigmond et al., 2016) has also shown significant forecast skill of more user relevant sea-ice metrics such as the first calendar date that sea ice melts (retreat date; Figs. 15a-c) or freezes up (advance date; Figs. 15d-f). Advance dates are skillfully predicted at lead times of 5 months on average (3.3 months for detrended anomalies), and retreat dates at lead times of 3 months (2.2 months for detrended





anomalies). For retreat dates, the main source of forecast skill is persistence, while advance date predictions benefit from predictable ocean temperatures.

Sea ice predictability is also assisted by persistence of sea-ice thickness (SIT), but CanSIPS does not take advantage of this
in that it currently employs an initialization method that uses only climatological SIT information. Since real time SIT observations are limited, Dirkson et al. (2015, 2017) have developed several statistical models of varying complexity for initializing SIT in operational predictions. These are based on predictors available in real time together with historical SIT values represented by the pan-Arctic Ice and Ocean Modelling System, or PIOMAS (Zhang and Rothrock, 2003), which is frequently used as a reference dataset for SIT due to the sparseness of historical SIT observations. The first such model
(known as "SMv1"), described in Dirkson et al. (2015), uses a statistical approach to find an optimal combination of sea ice concentration and sea level pressure information to provide useful sea ice thickness information. While this model reduces temporal- and spatial- mean absolute errors in the SIT initial conditions by 48% relative to the original CanSIPS initialization (when validated against PIOMAS SIT values), and shows consistent skill estimating ice volume in all months, much of this improvement in skill emerges from a more accurate representation of local negative trends in SIT. Two
additional statistical models, "SMv2" and "SMv3", that improve on SMv1 with respect to interannual variations in SIT anomalies are described in Dirkson et al. (2017), and seasonal sea ice volume from SMv3 is compared to that from CanSIPS initial conditions in Fig. 16.  Seasonal forecasting experiments using these SIT initial conditions demonstrate general improvement forecasting both pan-Arctic sea-ice extent and local sea-ice concentration compared to the current operational system, with most significant improvements afforded by initializing with either SMv2 or SMv3 (Dirkson et al., 2017).

**5 Conclusion**

We have assessed characteristics of snow, sea ice, and related climate parameters in Environment and Climate Change Canada's (ECCC) earth system model (ESM) CanESM2 and seasonal to interannual climate prediction system CanSIPS, with a focus on the Canadian sector of the Northern Hemisphere. This assessment is intended to provide a baseline for future versions of the models with respect to these important societally-relevant climate parameters. It has highlighted the
application of the Blended-5 multisource snow water equivalent (SWE) (Mudryk et al., 2015) and the CanESM2 Large Ensemble of climate simulations. In addition, it has highlighted new developments in sea ice, snow, and related climate parameter prediction on seasonal timescales. We summarize our key findings:

- The CanESM2 simulation of climate parameters over the Canadian land mass closely tied to snow – land surface temperature and precipitation on land in cold regions – lies well within the range of currently available international
models. There is considerable disagreement among observational datasets on the amount and geographical structure of snow water equivalent (SWE) in the satellite era. The CanESM2 simulation of SWE performs as well as available international models in this area. Even accounting for this observational uncertainty, however, there is a bias towards





excessive seasonal snow cover and unrealistic spatial distribution of SWE in the spring over the Canadian land mass and over the Northern Hemisphere as a whole. Excessive precipitation over the Canadian land mass contributes to this bias.

- Accounting for observational uncertainty, CanESM2 simulates a greater retreat of springtime snow over the satellite era than most of the available observations assessed here and other models that include large initial condition ensembles. The spatial pattern of the observed temperature, precipitation, and SWE trends is strongly influenced by internal variability. This makes it difficult to assess the model-simulated patterns of change in the variables we have examined. Nevertheless, Western Cordillera trends in SWE in CanESM2 represent a recent increase that is opposite to those found in typical CMIP5 models.

- Previously identified biases towards low Arctic sea ice extent are also reflected in regional biases: in Hudson's Bay and the Canadian Beaufort Sea sector, the sea ice extent is biased low and this undermines projections of when regional sectors of the Arctic will be ice free. In the current system, there are tradeoffs related to the resolution of geographical features in the CanESM and CanSIPS systems that impact both the snow and sea ice simulations. This provides an urgent area of improvement for future model development.

- Recent work suggests promising potential for seasonal forecasting of snow, sea ice and related climate parameters using CanSIPS. For example, accurate initialization of frozen and liquid soil water, in addition to improved SWE representation, might lead to significantly improved seasonal temperature forecasts. Furthermore, the Blended-5 example shows that accounting for observational uncertainty can lead to better understanding of forecast quality. This result suggests initialization could also be improved in this manner. This and related work has stimulated the development of ECCC's first experimental seasonal snow amount forecast product.

- Despite biases in the sea ice simulation, it is possible to develop potentially useful new seasonal forecast products for sea ice advance and retreat. In addition, implementing sea ice thickness initialization using indirect statistical predictors of thickness can improve sea ice forecasts compared to the current methodology. Motivated by the promising research results, improved sea ice thickness initialization (Lindsay et al., 2012) is being considered for implementation in the CanSIPS system.

Further improvements in the CanSIPS and CanESM climate prediction and projection capacity for snow, sea ice and related climate variables also hinge on assessing model process representation in more depth. For example, critical to capture accurately is the snow albedo feedback process, which governs the seasonality of snow cover and land surface temperature and hydroclimatic responses to climate change (Qu and Hall, 2007, 2014; Hall et al., 2008; Thackeray et al., 2015; Thackeray and Fletcher, 2016). Thackeray et al. (2015) show that CanESM2 places among the best CMIP5 models for all regions in terms of the overall simulation of snow cover fraction and snow-covered surface albedo. Further progress in this kind of process representation will be achieved in part through internationally coordinated inter-comparison efforts associated with CMIP6, including the Land Surface, Snow and Soil moisture Model Intercomparison Program (LS3MIP) and the Earth System Model Snow Inter-comparison Project (ESM-SnowMIP). Besides ongoing work on sea-ice mechanics





and its relationship, through wind driving, to sea ice drift, CanSISE research is also currently characterizing snow cover on sea ice in models and observations which also serves as a potential source of model error in the timing and amplitude of sea-ice growth and melt.

5    CanSISE demonstrates the utility of entraining a network of researchers bridging observational and modelling communities to focus on a related set of processes in evaluation of earth system models and climate prediction systems. The results suggest that there can be several benefits to updated multi-source observational datasets for climate prediction, monitoring, and assessment. Our focus in this paper has been on recently produced multi-source snow observational datasets, but our results strongly suggests the benefits of multi-source temperature, precipitation, and sea ice datasets following a similar
10   approach. We have articulated the tradeoffs involved in constraints on CanESM2's resolution in light of limitations of available advanced computing resources. Running the model at two degrees latitude/longitude permits the creation of the CanESM2 Large Ensemble set, but can entail under-resolution of key features of interest in applications, such as the Canadian Arctic Archipelago's channels and islands. We suggest that similar large ensembles be considered based on future model versions, accounting for these tradeoffs, and being complemented by ECCC regional climate model simulations (e.g.
15   Scinocca et al., 2016).



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



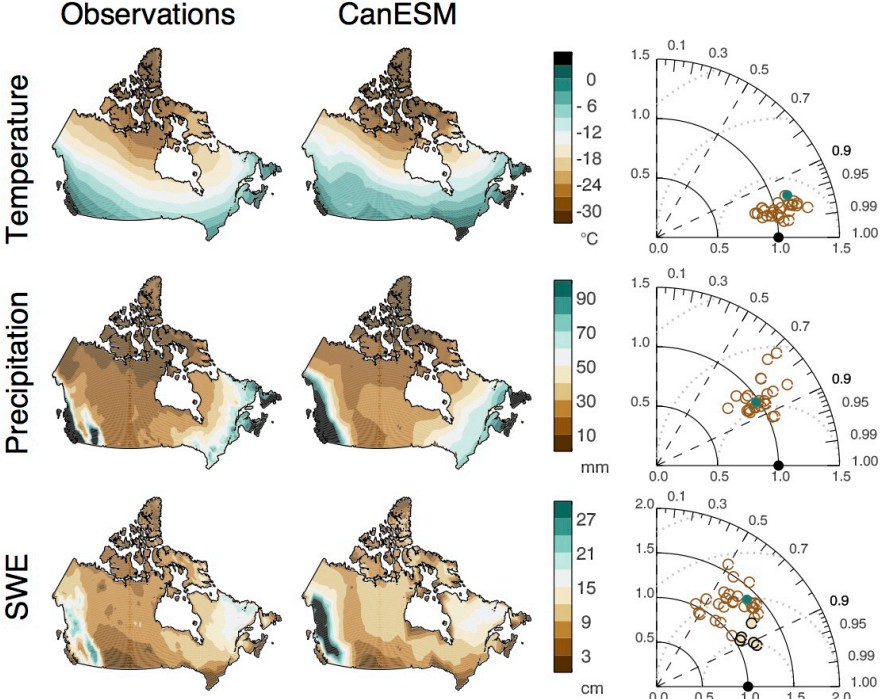

**Figure 1: Comparison of simulated Canadian climate in CanESM2 with observations and other climate models for 1981-2005. The left column shows observed January-March (JFM) mean temperature (top, HadCRUT4), precipitation (centre, CRU TS3.21) and snow water equivalent (bottom, Blended 5 data). The central column shows the same fields as simulated by the ensemble mean of the CanESM2 Large Ensemble. Observations and the CanESM2 output have been mapped to a common grid that represents the model grid spacing. The right column plots Taylor (2001) diagrams showing the correlation (related to the polar angle) and standard deviation relative to observations (distance from origin) of the patterns of these variables in observations (black dot), CanESM2 (green dot) and other CMIP5 climate models (brown circles). Models that are closer to the black dot representing the observations have smaller errors (standard error represented by dotted semicircles at intervals of 0.5). In the SWE Taylor diagram (bottom row), the filled light brown dots compare the individual Blended-5 datasets to the Blended-5 mean.**



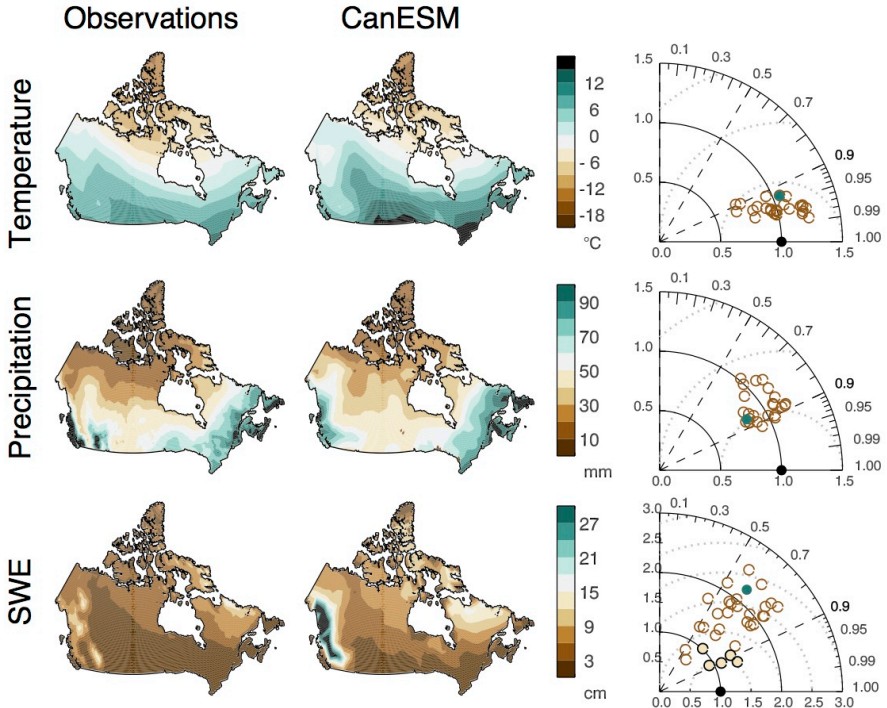

Figure 2: As in Fig. 1, for April-June (AMJ).





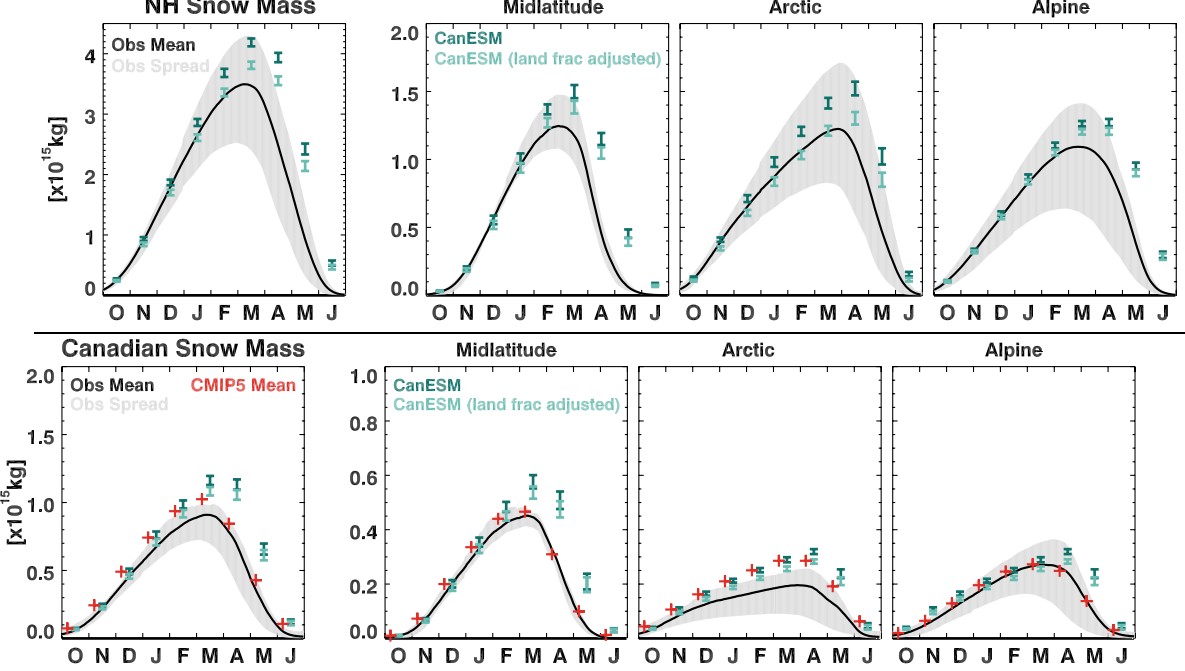

Figure 3: Top row: seasonal cycle of NH 1981-2005 snow mass (in 1015 kg) for the NH, midlatitude, Arctic, and alpine regions. Regions are defined as in Mudryk et al. 2015. Gray shading represents range of Blended-5 datasets, the black curve represents the Blended-5 mean, the dark teal points mark ensemble mean of CanESM2, and the light teal points mark CanESM2 adjusted to represent the same land fractions as the observational mask from the Blended-5 dataset. Bottom row: as in top row, but for Canadian land mass only, with CMIP5 model mean added in red.



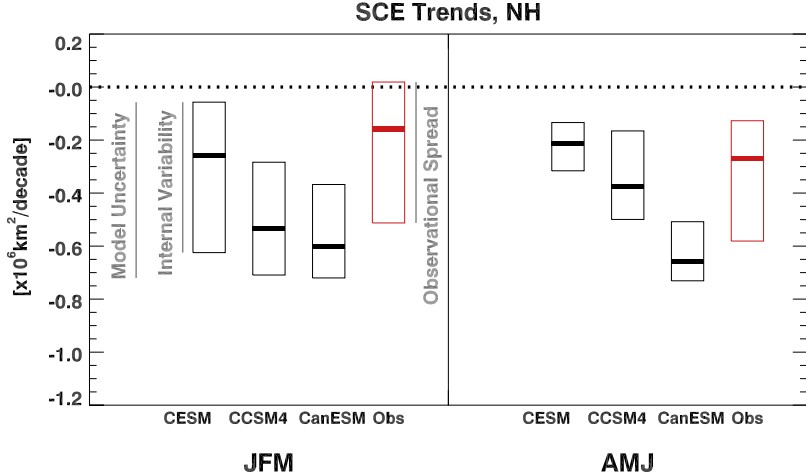

**Figure 4: Horizontal bars and boxes show the median and interquartile range of Northern Hemisphere snow cover extent trends calculated over 1981-2005 period for January-March (left) and April-June (right). Black boxes are used for trends calculated from models (large initial condition ensembles of NCAR CCSM4, NCAR CESM1 and CanESM2 as labelled), and red boxes are**

5  **used for trends derived from the Blended-5 SWE dataset (Mudryk et al. 2017). For the models, the boxes indicate the interquartile range (IQR) of trends captured in individual realizations of the ensembles. For the observations, the boxes indicate the IQR of observed trend estimates (which we note involves only five datasets over the historical record). Unlike the observational uncertainty, the uncertainty represented for the models is the impact of internal variability on estimated trends. Spread from each of these two sources along with additional spread due to model differences are indicated schematically by the extent of the vertical**

10  **grey lines. Note that the IQR of the observations is estimated from only five datasets; the IQR is much more robustly estimated by the 30-50 simulated realizations in the large ensembles.**



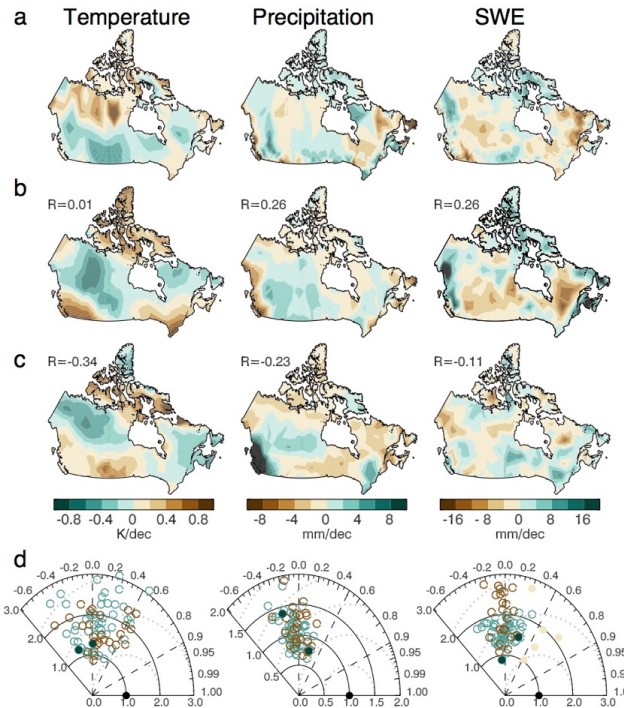

**Figure 5:** The top row shows the 1981-2005 trend in JFM temperature, precipitation, and SWE, with the spatial mean trend over Canada removed, based on the same datasets represented in Figs. 1-2. The bottom row shows the Taylor diagram of individual realizations of CanESM2 (teal) and CMIP5 (brown). The Taylor diagram for SWE also includes individual Blended-5 observations (light brown). The second row shows the temperature, precipitation, and SWE trends for a *best all-round match* case: a single realization having the greatest average pattern correlation across the three fields (temperature, precipitation, and SWE) and the two seasons of JFM and AMJ. The spatial pattern correlation coefficient of each field with its observational counterpart is labelled. The third row is the same as the second row but for a *worst all-round match* case: the single realization having the least (most negative) pattern correlation. The best and worst match cases are shown as filled green circles in the Taylor diagrams.

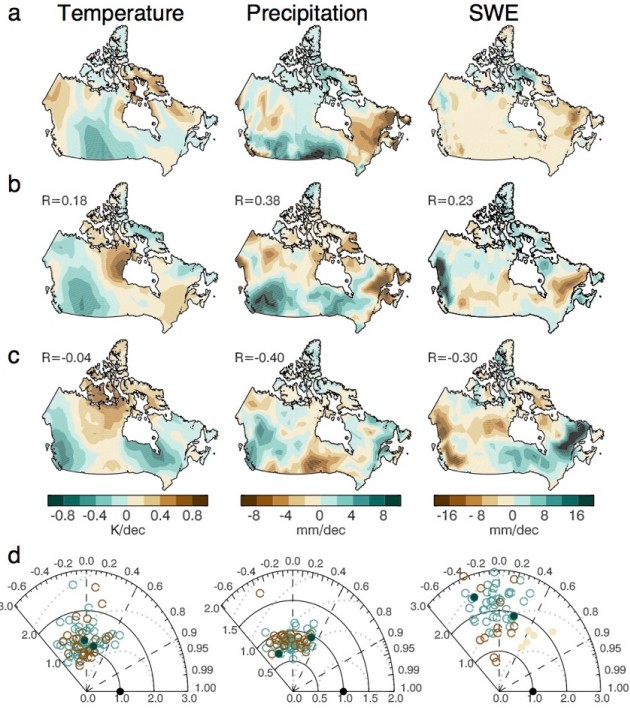

**Figure 6: A similar analysis to Fig. 5 but for AMJ. Left column and right column represent AMJ trends, and second and third columns represent the same best-match and worst-match realizations from Fig. 5, but with AMJ trends displayed.**



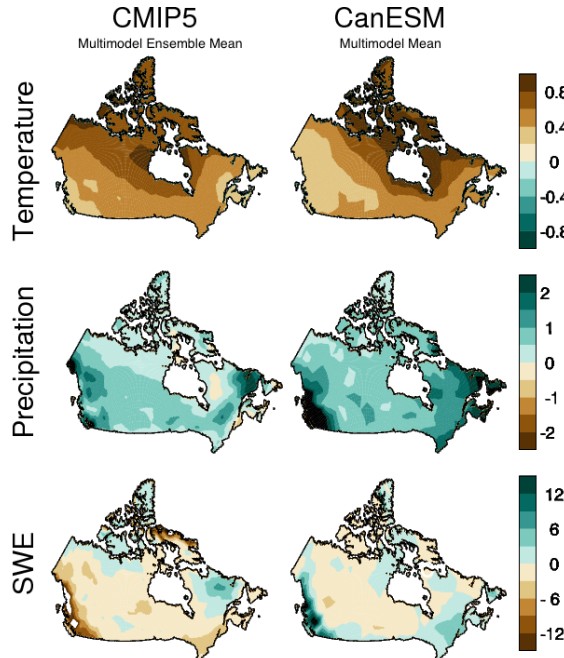

**Figure 7: The left column shows the multimodel mean of CMIP5 trends (without spatial mean removed) over the 1981-2005 historical period for JFM temperature, precipitation, and SWE. The right-hand column shows the corresponding multi-realization mean for the CanESM2 large ensemble. Unlike Figs. 5-6, the spatial mean of the trend has not been removed.**





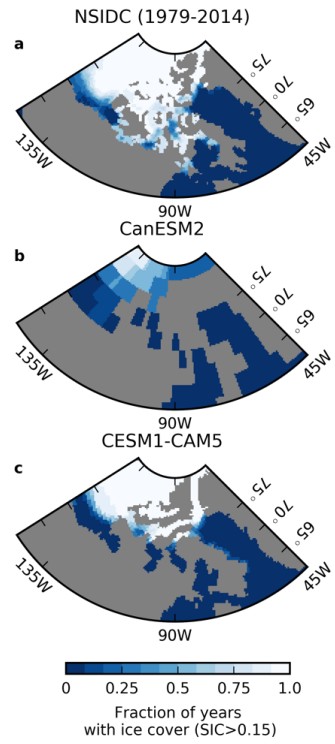

**Figure 8: a) Fraction of years (1979-2014) with September sea ice cover (sea ice concentration > 0.15) for the NSIDC passive microwave product on an EASE 25km grid. b) As in a) but for the CanESM2 model, remapped using a nearest neighbour remapping. c) As in b) but for the CESM1-CAM5 model. The grey shading indicates the land-sea mask for each dataset. The NCAR CESM1 CICE grid provide a rough indication of how geographical features of the Canadian Arctic, such as the Canadian Arctic Archipelago, might be resolved in future configurations of CanESM2. Note, however, that it is the ORCA1 grid that will be employed in the new CanESM2. Also evident is the low September sea ice extent bias in CanESM2.**



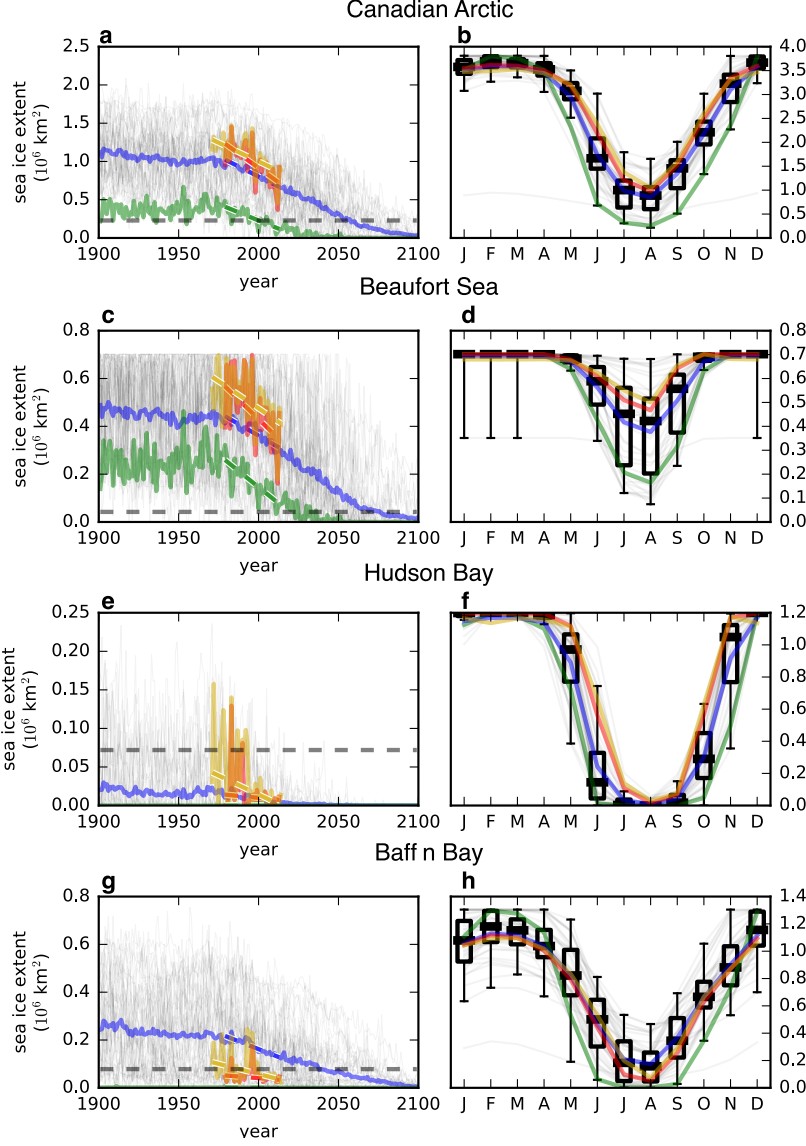

**Figure 9:** a): September sea ice extent (sea ice concentration larger than 0.15) for the Canadian Arctic [defined by the Canadian Ice Service Data Archive (CISDA) domain]. The CanESM2 model (green), CISDA (yellow), National Snow and Ice Data Center (NSIDC, red) and the multi-model mean (blue) are shown with their 1979-2013 trends. Individual models are shown in light gray. b: The Canadian Arctic seasonal cycle for 1979-2013. Box plots were added to describe the inter model spread (whiskers are 5th and 9th percentiles). c-d, e-f, g-h: same as a, b but for the Beaufort Sea, Baffin Bay and Hudson Bay, respectively. In panels e and g, the CanESM2 curve is close to zero. Sea ice amounts are scaled to account for the fraction of ocean present in the CanESM2 land sea mask (Laliberté et al. 2016).



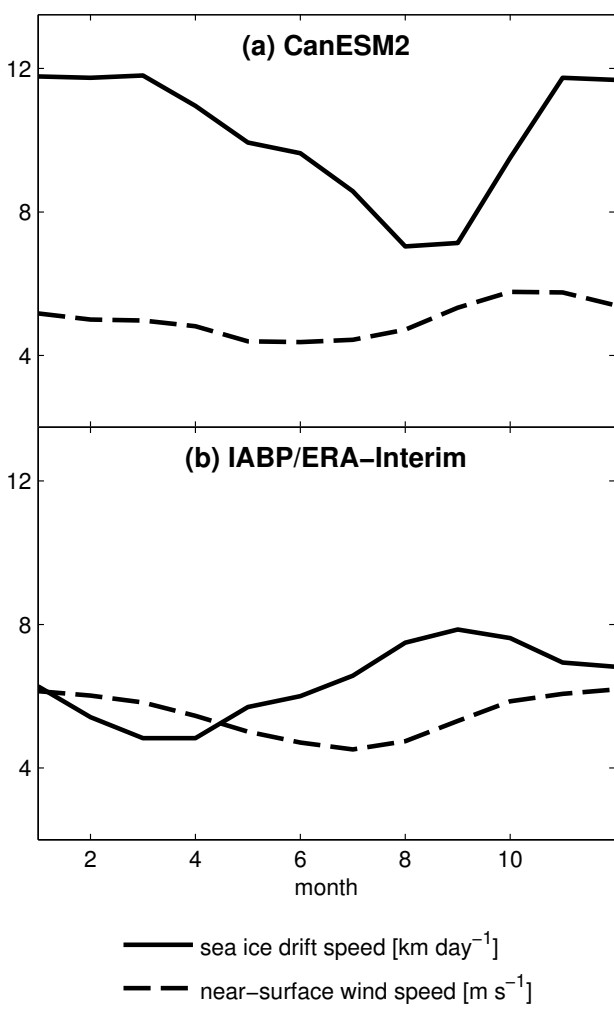

**Figure 10:** (a) Seasonal cycle of Arctic average sea ice drift speed (solid, in units of km day[-1]) and near-surface wind speed (dashed, in units of m s[-1]) from a historical run of CanESM2 averaged over 1979-2005. The spatial domain used for the calculation is the region north of 68°N for longitudes east of 103°E and west of 124°W and north of 79°N at all other longitudes, excluding gridpoints within 150 km of a coastline. This focuses on regions of year-round drifting ice, and excludes landfast ice. (b) As in (a) but using non-gridded drift speed measurements from the International Arctic Buoy Programme (solid; Tschudi et al., 2016) and near-surface wind data from ERA-Interim (dashed; Dee et al., 2011).



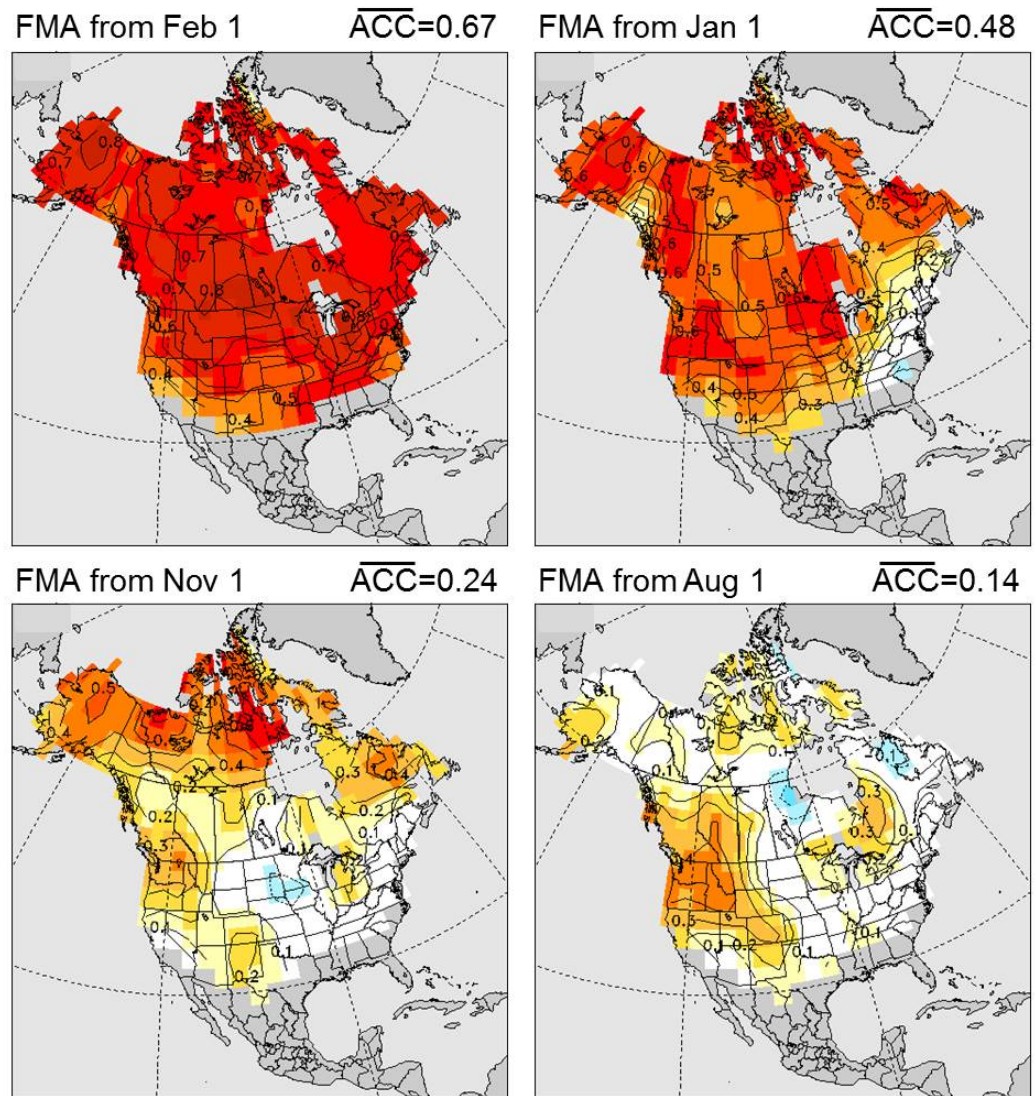

**Figure 11:** Anomaly correlation coefficient (ACC) skill for February-April (FMA) SWE of 1981-2010 CanSIPS hindcasts initialized at the start of February and the preceding January, November and August (lead times of 0, 1, 3 and 6 months respectively). The Blended-5 dataset of Mudryk et al. (2015) is used for observational verification. Contour interval 0.1, and the overbars denote spatial averages of ACC over areas of North America having seasonal snow cover.





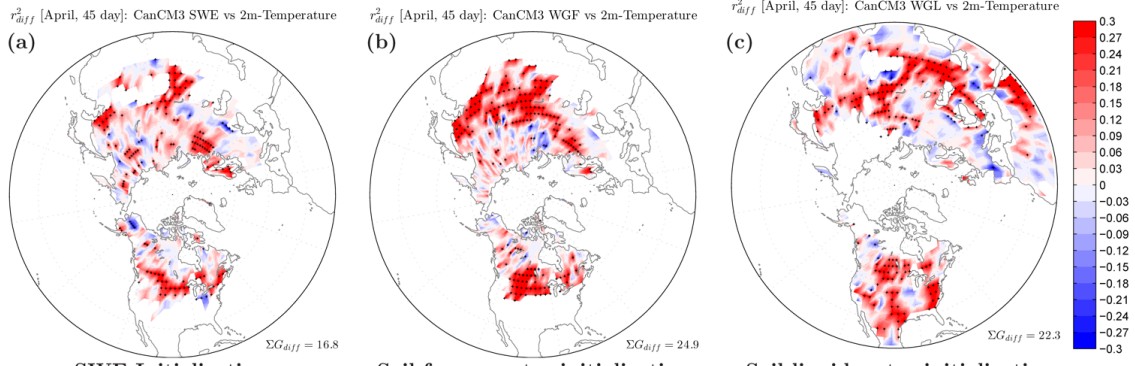

**Figure 12: Impact on the square of the anomaly correlation coefficient of initialization of SWE, and frozen and liquid soil water on springtime forecasts of surface air temperature at 45 day lead. Red colours indicate increase and blue colours decrease of the potential predictability (an idealized model based estimate of potential forecast skill). Based on Ambadan et al. 2015.**





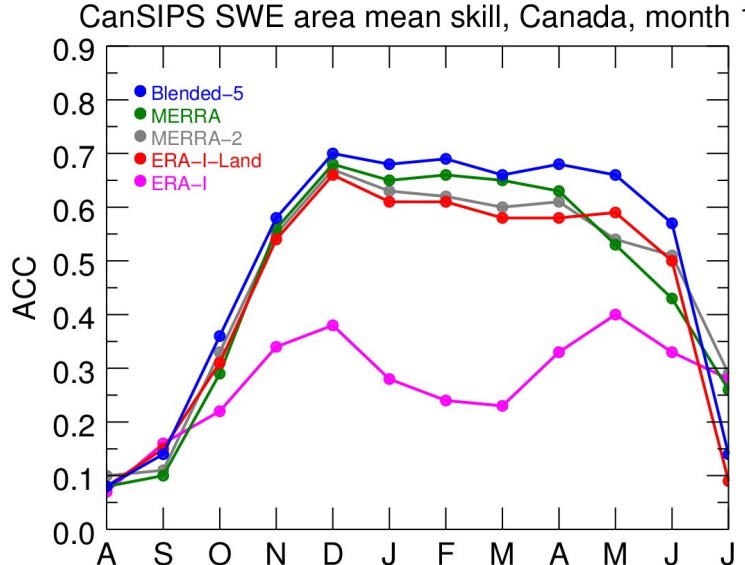

**Figure 13: Anomaly correlation coefficient (ACC) averaged over Canada for first-month CanSIPS forecasts of SWE, verified against several observation-based datasets including Blended-5 (blue), MERRA (green), MERRA-2 (grey), ERA-Interim Land (red), and ERA-Interim (magenta); the MERRA and ERA-Interim Land are components of Blended-5. Blended-5 shows the best agreement with the forecast, suggesting a strong influence of observational dataset on evaluation of forecast performance.**




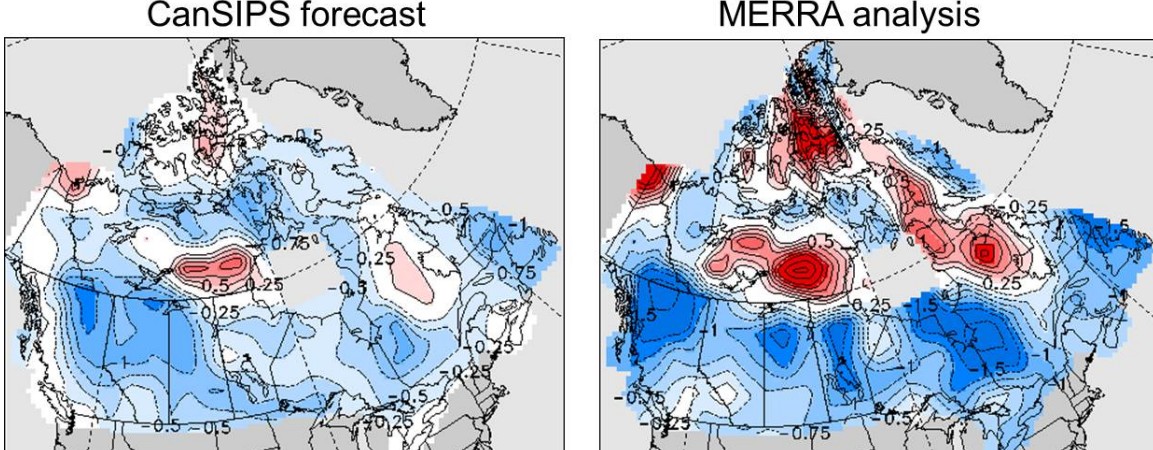

**Figure 14: Real time CanSIPS forecast of standardized anomalies of monthly-mean SWE for January 2016 (left), initialized at the end of the preceding month, compared to the MERRA analysis for the same period (right). Contour interval is 0.25, and anomalies are relative to a 1981-2010 base period.**





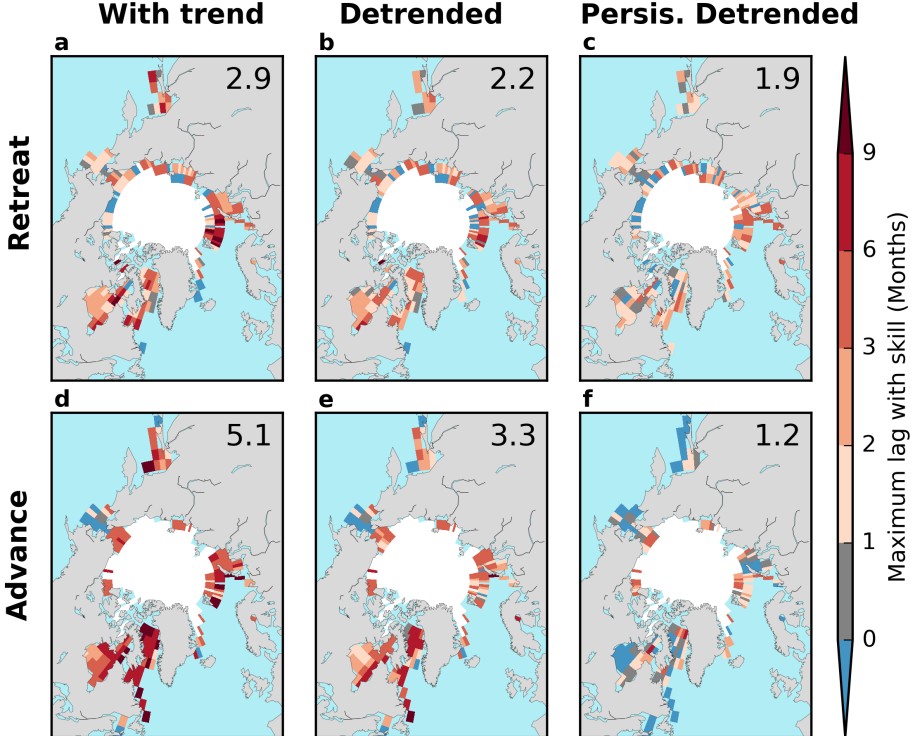

**Figure 15: Maximum lead time at which CanSIPS skillfully predicts retreat/advance dates (defined as the calendar date at which sea ice concentration first drops below/exceeds 50%), for total anomalies (first column), detrended anomalies (second column), and for a detrended persistence forecast based on persisting the observed initial sea-ice concentration anomaly. The numbers in the top**
5  **right corner of each panel indicate the Arctic average maximum lead time (in months). From Sigmond et al. (2016).**



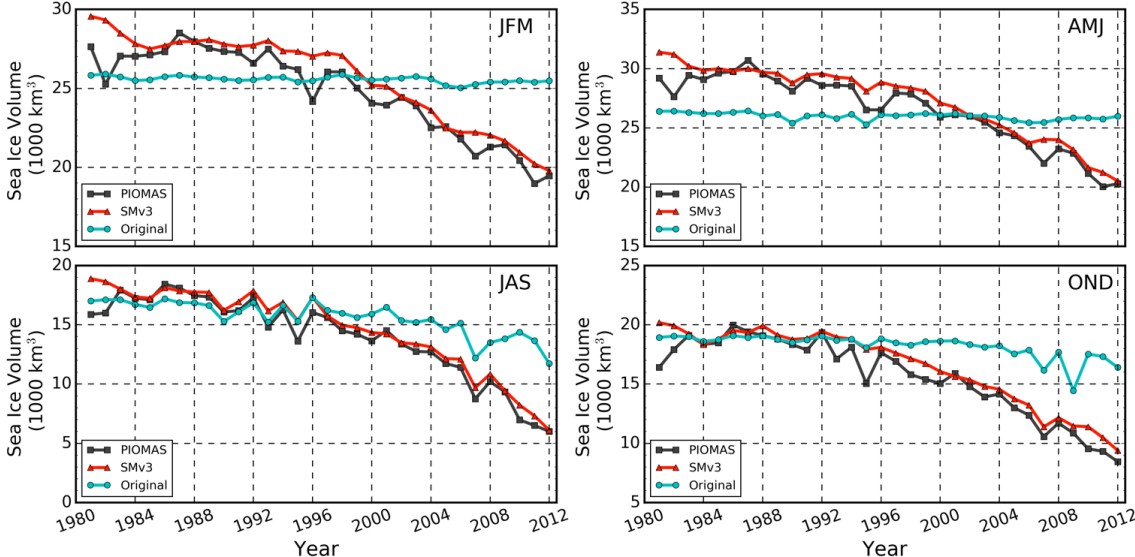

**Figure 16: Time series of seasonally averaged sea ice volume over the period 1981–2012, in units of $10^3$ km$^3$ for the PIOMAS sea ice thickness analysis (which assimilates observations) (black), the SMv3 statistical model of Dirkson et al. (2017) (red), and CanSIPS initial conditions (cyan). The CanSIPS system as originally configured incorporates no information about recent sea ice thinning and thus misses the recent downward trends of sea ice volume.**