# Peer review of "Assessment of Snow, Sea Ice, and Related Climate Processes in Canada's Earth-System Model and Climate Prediction System"

_The Cryosphere, 2017_

## Referee Comment (RC1) · R. L. H. Essery (Referee) · 13 Oct 2017

This is a well-written and useful record of how well snow and sea ice are simulated in an earth-system model and a related seasonal prediction system, albeit with rather brief discussions for the large number of figures shown.

There should at least be references to how the land surface components of CanSIPS and CanESM2 represent snow, and any differences between them.

Without a figure showing annual cycles in snow cover extent, it is hard to judge how (physically) significant the trends in Figure 4 are. It would also be nice to see some

time series of modelled and observed SCE to judge the trends and variability.

A reference could be added in the conclusions for the design of LS3MIP (https://www.geosci-model-dev.net/9/2809/2016/).

Figure 5: information that the IQR of observations is based on 5 datasets is repeated twice. Colour is redundant in this figure.

Figure 6: information that spatial means of trends have not been removed is repeated twice.

Figure 8: the latitude numbers could be rotated to be upright. Stating that the ORCA1 grid will be used without explanation or reference is not helpful.

Figures 11 and 14: contours are labelled, but colour bars would be helpful.

---

## Referee Comment (RC2) · Anonymous Referee #2 · 13 Nov 2017

Review of Assessment of Snow, Sea Ice, and Related Climate processes in Canada's Earth-System Model and Climate Prediction System by Paul Kushner and coauthors.

General comments

This draft is a review of studies carried out by the network of researchers within the project CanSISE. It deals with the quality of land snow and sea ice simulations in the CanESM2 earth system models, as well as the predictive capabilities of CanSIPS in land snow and sea ice predictions, with a focus on Canada and Canadian sector of the Arctic Ocean.

The draft is well written, and addresses a number of scientific studies, as well as

methodological topics related to model assessment. It will be an interesting and very useful reference for the Canadian ESM and its predictive capacities. I recommend this draft for publication. However, I did not find it easy to read and review, and I would like to raise a few points that, according to me, would improve the manuscript.

(i) I find the paper a bit unbalanced, the 'snow' discussion being more developed than the 'sea ice' one. Both discussions are also mostly unrelated. To me, it could question the usefulness of having both snow and sea ice discussed in a single paper. I think the authors should make an effort in harmonizing the presentation. It is obvious that the same parameters drive both snow and sea ice biases in CanESM, but it should be clearer. Additionally, the paper would be a bit clearer if sections 3 and 4 were structured with subsection on sea ice and snow (3a, 3b, 3c...).

(ii) Like other review papers, there is a need to find the optimal level of details. To me, there is a lot of references to past studies in this draft, which sometimes is not self-explanatory. Although it really makes me want to read papers written by CanSISE's partners, I would find it useful to have more details in a reference paper as a 'one-stop shop'. For instance, provide more details on datasets included in the Blended-5 SWE; remind the readers on some technical details: the components of CanESM/CanCM, ensemble generation in the large ensemble, in CanSIPS, the definition of the assessment regions... Tables could be used. Another related point is that the paper contains a large number of figures, which are not always well-discussed in the main text, and possibly too much in the captions.

(iii) Finally, as a non-Canadian (nearly) anonymous reviewer, I find the paper a bit too Canadian-centered. I am not surprised since it is a review paper from a Canadian project, and I acknowledge the major contribution of the CanSISE network to the field of snow and sea ice predictions. Though, the authors may wish to refer more to others' works...

Minor comments

Title The title should reflect that the assessment is on land snow, but it is clear in the abstract.

The abstract is clear and well-written.

1. Introduction

P2, L12, 'leading earth system': this is true, but a bit subjective. I would stay neutral and write 'global' earth system.

P2, L16, 'this study': this paper is more a review than a study.

P3, L1, 'related climate parameters': these parameters should be defined once in the draft: surface temperature, snow precipitation, sea surface temperature. . .

P3, L5, 'a more complete a characterization': too many a's.

P3, L5, 'observational uncertainty'.

2. Models and data used

This section should be a bit re-written. I would start by a description of the component of the coupled atmosphere-ocean-sea ice-land model, then describe the Earth System Model (ie with Carbon cycle), and finally explain what is CanSIPS. It is more in line with the order used to discuss the results in sections 3 and 4. And it seems to me more logical to describe the components, before describing the initialization method...

P3, L11-12: Merryfield et al. (2013a) refer to the multi-system sea ice predictions combining CanSIPS and CFSR. The reference should be Merryfield et al (2013b). Although there is an inversion of both references in the main text. . . which is the impression I have after reading the full draft...

P3, L16: what is 'it'?

P3, L19: It is not clear to me what (3) does exactly. Is it about calibration? It could be interesting to have an example.

P3, L23-29: Is it possible to provide any reference on the benefit of having an ESM? Is there any impact (positive or negative) of carbon cycle components on the state of the physical components?

P4, L1: CanESM alone includes a prognostic carbon cycle, isn't it?

P4, L3-6: Could the authors provide a bit more details on the reasons of improvements in CanCM4 relative to CanCM3? Is there any change in the model physics? Resolution?

P4, L19: compared to many other CMIP5 models and operational seasonal prediction systems (e.g. MetOffice is 1/4°, MeteoFrance is 1°...).

P4, L19-23: it would be interesting to have somewhere information about the size of ensembles run for seasonal predictions.

P4, L24-25, 'much higher resolution': what resolution?

P5, L7: what perturbations are used to generate the ensemble?

To me, in this section 2, a discussion on 'methodology' is missing. What is the motivation behind using the 'Large ensemble'? Maybe the words 'detection and attribution' should be written somewhere. How do the authors define the regions over which the assessment will be conducted? It would be useful at this stage, and will enable a discussion on the resolution of the land-sea mask for instance...

3. CanESM2 climatology and trends

P5, L15: is 'temperature' surface temperature? 2m-air temperature?

P5, L16: see above about the differences between CanCM4 and CanESM.

P5, L24: what about the orography in the (low resolution) model?

P7, L34 and sq: shouldn't it be the same for sea ice? See comment (i).

P8, L25: in terms of climatology

4. Snow and sea ice related forecast performance and development of CanSIPS

P9, L7-8: what is 'recently'? The review paper by Guémas and coauthors (see below) is never cited and provides a useful state-of-the-art of seasonal prediction of the Arctic sea ice.

P9, L26 and sq: the paragraph deals with land surface initialization, while it starts with reference to 'process representation of land surface'.

P10, L27: referring to Guémas et al (2016) would be fine here too.

P11, L4: a reference to Chevallier and Salas-Mélia (2012) seems relevant here.

5. Conclusions

P12, L9-13: does it mean that future developments of CanESM/CanSIPS include increase of resolution of the global model?

P12, L23: reference to Lindsay et al. (2012) not in reference list... Is it really a viable solution?

P12, L33-34: references for LS3MIP and ESM-SnowMIP?

Figures

Figure 1: what is 'temperature'?

Figure 3: definition of the regions considered (if I don't want to download Mudryk et al...)?

Figure 8: information on ORCA1° (not defined: reference?) seems not relevant here, or should be discussed in the main text (e.g. while presenting the components or their possible evolutions).

References

Chevallier, M., and Salas Y Mélia, D., 2012. The role of sea ice thickness distribution in
the Arctic sea ice potential predictability: a diagnostic approach with a coupled GCM. Journal of Climate, 25, 3025-3038, doi:10.1175/JCLI-D-11-00209.1

Guémas, V., Blanchard-Wrigglesworth, E., Chevallier, M., Day, J., Déqué, M., Doblas-Reyes, F., Fuckar, N., Germe, A., Hawkins, E., Keeley, S., Koenigk, T., Salas y Mélia, D., Tietsche, S., 2016. A review on Arctic sea ice predictability and prediction on seasonal-to-decadal timescales. Quarterly Journal of the Royal Meteorological Society, 142, 546–561. doi:10.1002/qj.2401.

Van den Hurk, B., Kim, H., Krinner, G., Seneviratne, S. I., Derksen, C., Oki, T., ... Viovy, N. (2016). LS3MIP (v1. 0) contribution to CMIP6: the Land Surface, Snow and Soil moisture Model Intercomparison Project–aims, setup and expected outcome, Geosci. Model Dev., 9, 2809–2832.

––––––––––––––––––––

---

## Author Comment (AC1) · 3 Feb 2018

General comment: Thank you for the feedback on this paper and the suggestions, which led to an improved manuscript. The major change suggested was to include more information on the snow cover extent (climatology), and we have accordingly revised Figure 3 to include observed and simulated SCE. A general point we will mention is that this overview of the system's performance was kept concise to establish a baseline for future model versions.

Specific comments/replies:

RC: This is a well-written and useful record of how well snow and sea ice are simulated in an earth-system model and a related seasonal prediction system, albeit with rather brief discussions for the large number of figures shown.

Reply: Thank you for the kind feedback. We attempted to keep the discussion concise with the aim of establishing a baseline record of this system's performance.

RC: There should at least be references to how the land surface components of CanSIPS and CanESM2 represent snow, and any differences between them.

Reply: Thank you for this suggestion. This is addressed in the revised text, where we have provided more detail on the component models in CanESM2, CanCM4, and CanCM3.

Changes to the manuscript: component models of CanESM2, CanCM4, and CanCM3 are listed and relevant citations included.

RC: Without a figure showing annual cycles in snow cover extent, it is hard to judge how (physically) significant the trends in Figure 4 are. It would also be nice to see some time series of modelled and observed SCE to judge the trends and variability.

Reply: Thank you for pointing this out. We have added two rows to Figure 3 showing the seasonal cycle of SCE and used this to calculate the relative size of the observed changes in SCE in the discussion of Figure 4. We decided not to add additional time series of SCE since these trends are discussed extensively in Mudryk et al. (2017) and Mudryk et al. (in review, https://www.the-cryosphere-discuss.net/tc-2017-198).

Changes to manuscript: Figure 3 (attached to this reply) expanded to include snow cover extent (SCE) and description of Figure 4 discusses magnitude of changes relative to climatology.

RC: A reference could be added in the conclusions for the design of LS3MIP (https://www.geosci-model-dev.net/9/2809/2016/).

Reply: Thank you, added.

Change to manuscript: Reference added.

RC: Figure 5: information that the IQR of observations is based on 5 datasets is repeated twice. Colour is redundant in this figure.

Reply: Thank you, corrected. We chose the coloring to distinguish the observations from the models.

Change to manuscript: Figure caption corrected.

RC: Figure 6: information that spatial means of trends have not been removed is repeated twice.

Reply: Thank you corrected.

Change to manuscript: Figure caption corrected.

RC: Figure 8: the latitude numbers could be rotated to be upright. Stating that the ORCA1 grid will be used without explanation or reference is not helpful.

Reply: Thank you, change to figure made and caption corrected.

Change to manuscript: Figure slightly modified and caption corrected.

RC: Figures 11 and 14: contours are labelled, but colour bars would be helpful.

Reply: Thank you, however in our view, the colors are intended to help distinguish features, but since contours are labelled and described in the caption we decided not to include a colorbar.

Changes to manuscript: none.
* * *
[Figure]

Figure 3: First row: seasonal cycle of NH 1981-2005 snow mass (in $10^{15}$ kg) for regions defined in Mudryk et al. 2015: midlatitudes – Northern Hemisphere nonalpine land regions south of 60°N, Arctic - nonalpine land regions north of 60°N, and alpine. Gray shading represents the range of Blended-5 datasets, the black curve represents the Blended-5 mean, the light teal points (in the first column of the first row only) mark the ensemble mean of CanESM2 using its land mask, and the dark teal points mark CanESM2 adjusted to represent the same land fractions as the observational mask from the Blended-5 dataset. The CMIP multi-model mean, adjusted to the observational mask, is show with red x symbols. The legend in the second column top row applies to the figure as a whole. Second row: as in top row, but for Canadian land mass only. Third and fourth rows are similar to the first and second rows, but for snow cover extent in $10^6$ km². The estimate of observed snow cover extent is derived from the Blended-5 SWE dataset using the approach of Mudryk et al. 2017, and is based on a 4 mm SWE threshold for the presence of snow cover; the simulated snow cover extent is based on snow cover fraction directly produced by the models.

22

**Fig. 1.** Revised Figure 3 for revised ms.

---

## Author Comment (AC2) · 4 Feb 2018

General comment: We thank the reviewer for the detailed review and helpful comments. In revising the ms we have made several changes that we hope address the overall issue of readability and other shortcomings identified. The most important change we have made is to revise the methodological description in Section 2 to make it easier to follow; we thank the reviewer for pointing out this particular shortcoming.

Before continuing, we wish to make two general points. First, snow and sea ice analysis were kept in the same paper to reflect work within the CanSISE project. Second, although there is some review material in this study to provide background, and one

figure included in the ms has been previously published, this assessment is not, in our view, a review paper. We thus did not feel it warranted to make this paper a 'one-stop shop' for information on the Canadian modelling systems and the observational work carried out previously, which would have increased the length and would have made the paper less readable. We'll return to these points below.

Specific comments and replies:

RC: This draft is a review of studies carried out by the network of researchers within the project CanSISE. It deals with the quality of land snow and sea ice simulations in the CanESM2 earth system models, as well as the predictive capabilities of CanSIPS in land snow and sea ice predictions, with a focus on Canada and Canadian sector of the Arctic Ocean.

The draft is well written, and addresses a number of scientific studies, as well as methodological topics related to model assessment. It will be an interesting and very useful reference for the Canadian ESM and its predictive capacities. I recommend this draft for publication. However, I did not find it easy to read and review, and I would like to raise a few points that, according to me, would improve the manuscript.

(i) I find the paper a bit unbalanced, the 'snow' discussion being more developed than the 'sea ice' one. Both discussions are also mostly unrelated. To me, it could question the usefulness of having both snow and sea ice discussed in a single paper. I think the authors should make an effort in harmonizing the presentation. It is obvious that the same parameters drive both snow and sea ice biases in CanESM, but it should be clearer. Additionally, the paper would be a bit clearer if sections 3 and 4 were structured with subsection on sea ice and snow (3a, 3b, 3c...).

Reply: As mentioned in the general comment above, the decision to keep snow and sea ice analysis in the same paper reflected the status of observational, modeling, and seasonal forecast work with this model and the CanSISE project. We placed more emphasis on the quality of the terrestrial snow simulation, and trend analysis with the

snow simulation, for two reasons: first, most of the observational work completed within the lifetime of CanSISE has focused on terrestrial snow; second, and a related point, is the fact that the sea ice simulation had well established shortcomings that simply can't be addressed with this version of the model which the project was tasked with analyzing. Despite differences in simulation quality of the free running model, both snow and sea ice seasonal prediction development has occurred within the scope of the CanSISE project, and we thought it was worth documenting both aspects within the current paper. As such, we have tried to establish a baseline for future assessment of the model.

The reviewer also suggests that it is obvious that the same parameters drive both snow and sea ice biases, but common sources for the important biases are not obvious to us. For example, the most unrealistic feature of the snow mass is a positive bias in spring - which appears to be related to a wintertime precipitation bias. The most unrealistic feature of the sea ice is a negative bias in summer, and this is likely contributed to by temperature biases but also issues with the sea ice and ocean simulation.

We agree that the sections needed to be better demarcated and hope that the current section headings make the presentation clearer.

Changes to ms: We have added subsections demarcating separate focuses on snow climatology and trends, and sea ice, in section 3. In the revision we have tried to better emphasize the distinctive biases in the snow and sea ice simulations. We have also provided more background on which results are better established from previous papers and which are original to this paper.

RC: (ii) Like other review papers, there is a need to find the optimal level of details. To me, there is a lot of references to past studies in this draft, which sometimes is not selfexplanatory. Although it really makes me want to read papers written by CanSISE's partners, I would find it useful to have more details in a reference paper as a 'one-stop shop'. For instance, provide more details on datasets included in the Blended-5 SWE;

remind the readers on some technical details: the components of CanESM/CanCM, ensemble generation in the large ensemble, in CanSIPS, the definition of the assessment regions. . . Tables could be used. Another related point is that the paper contains a large number of figures, which are not always well-discussed in the main text, and possibly too much in the captions.

Reply: We thank the reviewer for this comment and the suggestions. In the revision, we agree about the need for more details in the text regarding the makeup of CanCM3, CanCM4, and CanESM2. We have also tried to improve the description of the method for generating the large ensemble.

Beyond this, however, it does not seem like the best idea for us to repeat with tables etc. the many details that are found in previous papers. Although there is merit in the reviewer's suggestions, there is something of a judgement call to be made here regarding whether this paper should provide a 'one-stop shop' of the Canadian modelling systems as the reviewer requests.

This brings us to the reviewer's point about whether this study should be characterized as a 'review' or an 'assessment'. The ms, to us, reflects a record of where this particular modelling system sits in the context of its applications for climate prediction and projection, rather than a typical review paper.

Regarding the level of details of figure descriptions: We opted to keep the description of the figures concise but we have in this revision tried to add details where the text was unclear. We have in the case of Figures 5-6 moved some of the detail to the main text and in the case of Figure 3 and Figure 7 improved the description. Our general approach has been to put details in the caption that would be used to point out features to the reader that would have interrupted the flow of the main text.

Changes to the ms: We have added more details of the component model descriptions and methodological details. We have adjust the level of detail in the text and the captions to make the paper more readable.

RC: (iii) Finally, as a non-Canadian (nearly) anonymous reviewer, I find the paper a bit too Canadian-centered. I am not surprised since it is a review paper from a Canadian project, and I acknowledge the major contribution of the CanSISE network to the field of snow and sea ice predictions. Though, the authors may wish to refer more to others' works. . .

Reply: We thank the reviewer for this suggestion. We agree that the work needed to be placed in better context and have added some reference to previous work - the original ms did indeed leave out explicit mention of work going on in seasonal sea ice prediction starting from 2011. Our focus on Canada and the Canadian Arctic did to some extent reflect the focus of research within this project and less of a broad assessment on the US and Eurasian cryosphere. We believe that the focus is similar to other regionally focused literature in this and other international journals, and reflects the regional focus of individual research programs. In addition, the seasonal prediction system results need to be focused on Canadian regions where products have been developed and assessed (e.g. the seasonal snow depth product in Fig. 14). Thus we have opted not to change the focus.

Changes to ms: Citations and references to previous work have been added, including reviewer's suggestions and others we thought suitable.

RC: Minor comments

The abstract is clear and well-written

P2, L12, 'leading earth system': this is true, but a bit subjective. I would stay neutral and write 'global' earth system.

Reply: Thank you, language adjusted.

Changes to ms: Text has been changed.

RC: P2, L16, 'this study': this paper is more a review than a study.

Reply: Thank you, as we mentioned above, we are not in full agreement with the reviewer about the characterization of this work, but we have reworded this.

Change to ms: Text has been changed to say "The purpose of this paper is to evaluate the ability ..."

RC: P3, L1, 'related climate parameters': these parameters should be defined once in the draft: surface temperature, snow precipitation, sea surface temperature. . .

Reply: Thank you, language adjusted, we now say "This study focuses on snow, sea ice and related climate parameters and processes...". Please note that information related to surface winds (Figure 10), soil moisture content (Fig. 12), and sea level pressure (implicit in the work generating Fig. 16) is involved in the analysis here, but we didn't want to include a long list at this point.

Changes to ms: Text has been changed.

RC:

P3, L5, 'a more complete a characterization': too many a's.

P3, L5, 'observational uncertainty'.

Reply: Thank you, corrected.

Changes to ms: corrections made.

RC: 2. Models and data used This section should be a bit re-written. I would start by a description of the component of the coupled atmosphere-ocean-sea ice-land model, then describe the Earth System Model (ie with Carbon cycle), and finally explain what is CanSIPS. It is more in line with the order used to discuss the results in sections 3 and 4. And it seems to me more logical to describe the components, before describing the initialization method...

Reply: We have revised the order of the description according to the reviewer's suggestions, and thank the reviewer for this suggestion to improve the text. We opted not to include a lot of detail as mentioned above, but have been more explicit about CanESM2, CanCM3, CanCM4, and why we did not carry out a thorough analysis of CanCM4 in this paper.

Changes made to ms: Section 2 has been largely rewritten to reflect the reviewers comments and provide a clearer path to the subsequent analysis.

RC:

P3, L11-12: Merryfield et al. (2013a) refer to the multi-system sea ice predictions combining CanSIPS and CFSR. The reference should be Merryfield et al (2013b). Although there is an inversion of both references in the main text. . . which is the impression I have after reading the full draft...

P3, L16: what is 'it'?

Reply: Thank you, both points corrected.

Changes made to ms: References and text corrected.

RC: P3, L19: It is not clear to me what (3) does exactly. Is it about calibration? It could be interesting to have an example.

Reply: This point was to make the broad point that it is an operation product that is used for a variety of purposes - a couple of illustrative examples are included.

Changes made to ms: a reference to a figure in the paper the CanSIPS web site have been added.

RC: P3, L23-29: Is it possible to provide any reference on the benefit of having an ESM? Is there any impact (positive or negative) of carbon cycle components on the state of the physical components?

Reply: It is true that Merryfield et al. 2013a included a limited assessment of CanCM3

and CanCM4, which in principle could have been carried out in the areas of snow and sea ice for this paper. However, the Earth System Model CanESM2 is the only ECCC model used in CMIP5, that is forced with historical and projected forcings. Thus, the relevant runs of CanCM3 and CanCM4 under CMIP5 forcing protocols are not available for a clean comparison of the kind suggested by the reviewer. In the absence of consistently forced simulations, it would be difficult to attribute specific changes to the carbon cycle or to other differences. We have added more background explaining this in the text.

In addition, the reviewer's comment did remind us of a salient point that springtime positive biases in SWE were also found in the CanSIPS CanCM3 and CanCM4, whether or not they were constrained by observed meteorological data, in Sospedra-Alfonso et al. (2016b). A point to this effect has been added in Section 3.1

Changes made to ms: The reasons for the focus on CanESM2 has been clarified, and more details have been provided to explain the relationships between the different model systems.

RC: P4, L1: CanESM alone includes a prognostic carbon cycle, isn't it?

Reply: Thank you, the revised text has corrected this.

Changes to ms: this has been corrected in the revision as part of the larger rewrite of section 2.

RC: P4, L3-6: Could the authors provide a bit more details on the reasons of improvements in CanCM4 relative to CanCM3? Is there any change in the model physics? Resolution?

Reply: Thank you for the suggestion, we mention the changes but leave the details to the description in Merryfield et al. (2013a).

Changes to ms: We have added some more information about CanCM3 and CanCM4 in the larger rewrite of section 2.

RC: P4, L19: compared to many other CMIP5 models and operational seasonal prediction systems (e.g. MetOffice is 1/4âŮę , MeteoFrance is 1âŮę ...).

P4, L19-23: it would be interesting to have somewhere information about the size of ensembles run for seasonal predictions.

P4, L24-25, 'much higher resolution': what resolution?

Reply: Thank you, the suggested changes have been made.

Changes made to ms: several changes.

RC: P5, L7: what perturbations are used to generate the ensemble?

Reply: As described in Merryfield et al. 2013, each ensemble member, assimilates the same data but originates from different initial conditions. Thank you for the suggestion, but we opted not to include this level of detail in the current paper, since it requires explaining more about the initialization with details that are documented in the Merryfield paper.

RC: To me, in this section 2, a discussion on 'methodology' is missing. What is the motivation behind using the 'Large ensemble'? Maybe the words 'detection and attribution' should be written somewhere. How do the authors define the regions over which the assessment will be conducted? It would be useful at this stage, and will enable a discussion on the resolution of the land-sea mask for instance...

Reply: Several modifications to the text were made, although the land-sea mask is not referred to in this section.

Changes made to ms: text changed to clarify motivation of using large ensemble and attribution of observed trends to externally forced signals.

RC:

P5, L15: is 'temperature' surface temperature? 2m-air temperature?

Reply: Thank you, this corrected to land-surface temperature throughout ms (as per documentation of HadCRUT4 dataset).

Changes made to ms: "temperature" changed to land-surface temperature in most places.

RC:

P5, L16: see above about the differences between CanCM4 and CanESM.

Reply: This has been changed in the rewrite of Section 2.

RC: P5, L24: what about the orography in the (low resolution) model?

Reply: We are not sure what the reviewer means by this. The orography is resolved as per the resolution of the grid. Moving to higher resolution would potentially improve some details over the Western Cordillera. But note that the precipitation bias in Figure 1 is widespread and not confined to the Western Cordillera only.

RC: P7, L34 and sq: shouldn't it be the same for sea ice? See comment (i).

Reply: The reviewer is likely correct, but the point in response to (i) is that a detailed regional look at sea ice trends for this model is problematic because of the pronounced biases in the sea ice climatology.

RC: P8, L25: in terms of climatology

P9, L7-8: what is 'recently'? The review paper by Guémas and coauthors (see below) is never cited and provides a useful state-of-the-art of seasonal prediction of the Arctic sea ice.

Reply: thank you, corrections, clarification, and additional references were carried out.

Changes to the ms: minor corrections carried out.

RC: P9, L26 and sq: the paragraph deals with land surface initialization, while it starts with reference to 'process representation of land surface'.

Reply: Added the words "and initialization" here. But we feel that the term 'process' should still be used. The point is that although no process sensitivity studies were carried out, distinctively initializing different components of the land surface model and looking at the related potential predictability provides insight into processes relevant to the prediction problem.

Changes made to ms: minor corrections.

RC: P10, L27: referring to Guémas et al (2016) would be fine here too.

P11, L4: a reference to Chevallier and Salas-Mélia (2012) seems relevant here.

P12, L9-13: does it mean that future developments of CanESM/CanSIPS include increase of resolution of the global model?

Reply: Thank you, references added. We did not want to comment on future releases of the model beyond the points made in the ms.

Changes made to ms: references added.

RC: P12, L23: reference to Lindsay et al. (2012) not in reference list. . . Is it really a viable solution?

Reply: Thanks for pointing this out. Thickness initialization is more of an aspiration, so we added a qualifier and another reference.

Changes made to ms: minor changes on this line.

P12, L33-34: references for LS3MIP and ESM-SnowMIP?

Reply: Thank you, added reference for LS3MIP but are not aware of an ESM-SnowMIP paper in the peer-reviewed literature, and opted not to include a link to the ESM-SnowMIP project.

Changes made to ms: reference added.

RC: Figure 1: what is 'temperature'?

Figure 3: definition of the regions considered (if I don't want to download Mudryk et al...)?

Figure 8: information on ORCA1âǓę (not defined: reference?) seems not relevant here, or should be discussed in the main text (e.g. while presenting the components or their possible evolutions).

Reply: Thank you, all these changes have been made.

Changes made to ms: Figure captions changed.

RC: References Chevallier, M., and Salas Y Mélia, D., 2012. The role of sea ice thickness distribution in the Arctic sea ice potential predictability: a diagnostic approach with a coupled GCM. Journal of Climate, 25, 3025-3038, doi:10.1175/JCLI-D-11-00209.1 Guémas, V., Blanchard-Wrigglesworth, E., Chevallier, M., Day, J., Déqué, M., Doblas-Reyes, F., Fuckar, N., Germe, A., Hawkins, E., Keeley, S., Koenigk, T., Salas y Mélia, D., Tietsche, S., 2016. A review on Arctic sea ice predictability and prediction on seasonalto-decadal timescales. Quarterly Journal of the Royal Meteorological Society, 142, 546–561. doi:10.1002/qj.2401. Van den Hurk, B., Kim, H., Krinner, G., Seneviratne, S. I., Derksen, C., Oki, T., ... Viovy, N. (2016). LS3MIP (v1. 0) contribution to CMIP6: the Land Surface, Snow and Soil moisture Model Intercomparison Project–aims, setup and expected outcome, Geosci. Model Dev., 9, 2809–2832.

Reply: Thank you for these references.

Changes made to ms: citations and references added.